# Expression and Functional Analysis of the Smo Protein in *Apis mellifera*

**DOI:** 10.3390/insects15070555

**Published:** 2024-07-22

**Authors:** Lina Guo, Jue Wang, Diandian Yu, Yu Zhang, Huiman Zhang, Yuan Guo

**Affiliations:** 1College of Animal Science, Shanxi Agricultural University, Jinzhong 030801, China; linaguo@126.com (L.G.); wangjueoran@163.com (J.W.); yudiandian0121@163.com (D.Y.); 15183537865@163.com (Y.Z.); m19915323437@163.com (H.Z.); 2College of Horticulture, Shanxi Agricultural University, Taiyuan 030031, China

**Keywords:** Smo, *Apis mellifera*, hedgehog, olfactory receptor, functional analysis

## Abstract

**Simple Summary:**

Honeybees (*Apis mellifera*) rely on their olfaction for various daily activities and tasks. Smoothened (Smo) serves as a critical element in the Hedgehog pathway and is likely associated with the modulation of the olfactory capabilities of bees. This study involved the identification, localization, and functional validation of Smo. The findings establish a foundational understanding of the role of Smo in influencing the olfactory abilities of bees.

**Abstract:**

Smoothened (*Smo*) is a critical component regulating the Hedgehog signaling pathway. However, whether *Smo* is associated with the modulation of olfactory recognition capabilities of bees remains unclear. In this study, we amplified *Smo* from *Apis mellifera.* The coding sequence of *Smo* was 2952 bp long, encoded 983 amino acids. *Smo* was most highly expressed in the antennae. Cyclopamine (200 μg/mL) significantly reduced but purmorphamine (800 μg/mL) significantly increased *Smo* expression (*p* < 0.05). *OR152* and *OR2* expression in the cyclopamine group significantly decreased, whereas *OR152* expression in the purmorphamine group significantly increased (*p* < 0.05). A significant decrease in the relative values of electroantennography was observed in the cyclopamine group exposed to neral. Behavioral tests indicated a significant decrease in the attractive rates of neral, VUAA1, linalool, and methyl heptenone in the cyclopamine group. Conversely, the selection rates of linalool and methyl heptenone in the purmorphamine group significantly increased. Our findings indicate that Smo may play a role in modulating olfactory receptors in bees.

## 1. Introduction

The Hedgehog (Hh) signaling pathway, initially discovered in *Drosophila* [1], is a key regulator of embryonic development and tissue homeostasis [2]. The pathway is highly evolutionarily conserved [3], performing crucial roles in both vertebrates and invertebrates [4]. These roles include the regeneration of tissue, such as bone [5], neural tissue [6,7,8], skin, muscle [9] tissue, and the gastrointestinal mucosa [10]. Disruptions in the signal transduction of the Hh pathway can lead to the formation of tumors, such as basal cell carcinoma and neuroblastoma [11]. The immune system activates the Hh signaling pathway after injury [12,13,14].

Smoothened (Smo) is a critical component regulating the Hh signaling pathway [15]. The typical Hh pathway involves a multi-component signaling cascade, requiring a complex between Hh and the Smo receptor mediated by Patched (PTC), ultimately leading to the expression of the downstream factors Gli and cubitus interruptus (Ci) [16]. The Smo receptor, homologous to G-protein-coupled receptors, comprises a single peptide chain with seven transmembrane domains. The N-terminus is extracellular, whereas the C-terminus is intracellular, with highly conserved amino acid sequences in the transmembrane region. The serine and threonine residues at the C-terminus act as phosphorylation sites, facilitating the binding of phosphate groups during protein kinase catalysis [17]. Thus, elucidating its fundamental information and mechanism of action can provide a theoretical foundation for understanding its role in the Hh signaling pathway in *Apis mellifera*.

Bees depend on their olfactory system for foraging, identification of oviposition sites, communication, and reproduction [18,19]. The recognition of external odor molecules by insects is a highly intricate process involving various chemosensory proteins, with olfactory receptors (ORs) playing a pivotal role as key olfactory sensory proteins. Olfactory receptors are classified into two types, generalist and specialist. The generalist reacts to many kinds of odors and the specialist reacts to only one kind of odor, such as sex pheromone receptors [20,21].

Insect ORs can be categorized into conventional ORs and atypical ORs (Orco receptor family). Conventional ORs show low homology among different insect species, indicating high diversity, whereas the Orco receptor family exhibits high conservation across different insect species [22]. The Hh signaling pathway regulates the expression of ORs. In mammals, the Hh family member Sonic hedgehog is present in human nasal mucus and the dendrites of olfactory sensory neurons in the olfactory bulb of the rat brain [23]. A decrease in Sonic hedgehog levels in human nasal mucus is associated with olfactory loss [24,25]. Knocking down the *Smo* gene or inhibiting *Smo* expression with cyclopamine in mice affects the transport of ORs [26]. Sanchez et al. [27] demonstrated that the Hh signaling pathway also regulates the odor response of adult *Drosophila* olfactory neurons. However, to the best of our knowledge, there have been no reports on the functional analysis of the Smo protein in *A. mellifera*.

In this study, we analyzed the sequence of the Smo protein in *A. mellifera*, explored its expression characteristics in various tissues using quantitative reverse transcription polymerase chain reaction (qRT-PCR), observed its localization and expression in the antennae via in situ hybridization, and employed cyclopamine and purmorphamine to investigate whether the inhibition or activation of Smo activity would regulate the expression of ORs. Additionally, electroantennography (EAG) and a Y-shaped olfactometer were used for further functional validation.

## 2. Materials and Methods

### 2.1. Sample Collection

*A. mellifera* individuals were collected from the experimental base of the College of Animal Science at Shanxi Agricultural University. Antennae, heads (excluding antennae and proboscis), thoraxes, abdomens, legs, wings, and whole tissues of multiple worker bees were collected in quantities of 100 mg. The tissues were kept in liquid nitrogen during the isolation process and stored at −80 °C overnight for subsequent use.

### 2.2. Drug Feeding

*A. mellifera* were fed cyclopamine (APE×BIO, Houston, TX, USA) and purmorphamine (APE×BIO, Houston, TX, USA) using a drug concentration gradient of 0, 200, 500, and 800 μg/mL. Bees were captured and placed in homemade feeding boxes, with each box containing 20 individuals, and were fed for 24 and 48 h, resulting in four groups: 24 h cyclopamine, 48 h cyclopamine, 24 h purmorphamine, and 48 h purmorphamine. Next, 20 μL of the drug was transferred into 2 mL of sugar water, which was shaken thoroughly on a shaker to mix and injected into the enzyme label plate in the feeding boxes. Bees were then captured for further study. The feeding boxes were placed in an incubator (XT5107 Humidity Incubators, Ningbo, China) at 30 ± 2 °C and 65 ± 5% humidity.

### 2.3. Total RNA Extraction and cDNA Synthesis

For RNA extraction, we extracted RNA from whole bees for the gene identification part; we extracted RNA from different tissues of *A. mellifera* separately for different tissue expression; and we extracted RNA from the antennae and thorax of *A. mellifera* after feeding purmorphamine and cyclopamine according to the results of different tissue expression. Total RNA was extracted using TRIzol, and RNA quality was assessed via agarose gel electrophoresis. Following the instructions of the reverse transcription kit (Takara, Beijing, China), RNA was reverse transcribed into cDNA. The synthesized cDNA was stored at −20 °C for later use.

### 2.4. Smo Gene Amplification

Utilizing the GenBank reference sequence of *A. mellifera Smo* (XM_395373.6), we amplified it in three segments by designing three primers, i.e., P1, P2, and P3; nestle primers were labeled using horizontal lines. The primers were synthesized by Sangon Biotech (Shanghai, China) Co., Ltd. The PCR products were subjected to 1% agarose gel electrophoresis, and the desired fragments were purified after gel recovery. The purified fragments were sent to Sangon Biotech (Shanghai) Co., Ltd. for sequencing, and the correct sequences were submitted to GenBank (OP616714). Specific primers for qRT-PCR were then devised based on the cloned *Smo* gene sequence and the reference gene *Arp1* (NM_001185145.1). The primer sequences are detailed in Table 1.

### 2.5. Expression of Smo

qRT-PCR was employed to detect the expression of *Smo*, *OR151*, *OR152*, and *OR2* in *A. mellifera* (primers in Table 1). The reference gene is *Arp1*. RNA extraction and cDNA reverse transcription were performed according to Section 2.3. Each sample was run in triplicate. The 2^−△△Ct^ method was used to analyze the sequences based on standard and fluorescence curves.

### 2.6. Localization of Smo Genes in the Antennae

The *A. mellifera* antennae were placed on the stage, the antennae were embedded by OCT (SAKURA, Tokyo, Japan), adhered to the surface of the carrier tray by freezing, and the thickness of the slice was about 5 μm. Then, it was fixed with 4% paraformaldehyde, washed fully with PBS, and the blocking solution was added for 1 h; the blocking solution was then shaken off, the primary antibody (Zoonbio Biotechnology Co., Ltd., Nanjing, China), which was diluted 500 times, was added, and the wet box was placed at 4 °C overnight. The negative control group was washed with PBS for 5 min × 3 times, the washing solution was shaken off, the diluted 500-fold secondary antibody was added, incubated at 25 ± 1 °C in the dark for 1 h, and then washed with PBS for 5 min × 3 times. Then, the residual liquid on the section was blotted dry with absorbent paper, the fluorescent quencher solution was added dropwise on the section, the film was covered, the coverslip was covered to avoid bubbles, and the fluorescence microscope (Leica, Wetalar, Germany) was used to take pictures and records under the blue excitation light.

### 2.7. EAG Recording

The test honeybees were treated with purmorphamine and cyclopamine, respectively, as outlined in Section 2.2. The groups demonstrating optimal feeding effects in terms of concentration and duration were chosen for antennal electrophysiology analysis. Standardized compounds were referenced to those reported by Claudianos et al. [28].

The antennae were carefully cut at the base and placed into EAG electrode probes (Syntech, Hilversum, The Netherlands) with a drop of Spectra 360 electrode gel (Parker Lab, Inc., Fairfield, NJ, USA). The electrodes were connected to a DC/AC amplifier (Syntech IDAC-4) and a stimulation amplifier (Syntech Combi Probe 10 ×) connected to a computer. A stimulation gas control device (Syntech CS-55) was used to pass gas over the antennae through a Pasteur pipette for 0.5 s. A stimulus flow of 40 mL/min carried the odor of the sample. The constant flow rate was 500 mL/min and the gas was filtered and humidified.

With each stimulus, 10 µL of the test solution was pipetted onto a fresh 5 × 3 cm strip of filter paper and liquid paraffin was used as the reference check. To reduce error and allow the antennae to recover, each antenna was stimulated twice with each substance at 30 s intervals. Three repetitions were conducted for each tested sample on different antennae, and the antennae were prepared immediately before use to maintain their activity. The standard compounds are listed in Table 2.

The antennal electrophysiological response for compounds was calculated as follows:RV EAG = (Vs − Vbmean)/Vbmean
where RV EAG represents the relative EAG response value, Vs denotes the sample response value, and Vbmean is the mean response value of the control. This formula quantifies the relative alteration in antennal electrophysiological response compared to that in the control and is expressed as a ratio.

### 2.8. Behavioral Tests

The Y-shaped olfactometer (Nanjing Possum Instrument Co., Ltd., Nanjing, China) (stem 20 cm, arm 15 cm, forming a 75° angle) was used to investigate the behavioral responses. To ensure the accuracy of the experimental results, the entire procedure was conducted in a darkroom. Approximately 10 μL each of the sample and liquid paraffin (control) were separately applied to two filter papers measuring 3 cm in length and 1.5 cm in width, which were then placed in the odor source bottle. The gas flow rate was set at 300 mL/min. Groups with selected concentrations and durations were fed accordingly and then removed from the constant temperature and humidity chamber and placed in a darkroom. The indoor temperature was maintained at 23 ± 2 °C, and the testing period was from 8 a.m. to 4 p.m. Prior to the experiment, the gas pump was turned on for 1 min to allow the gas to pass through the olfactometer. Subsequently, the experimental honeybees were placed in the middle of the olfactometer, and their selective responses to different odors were observed and recorded.

The assessment criteria were as follows: each experimental bee’s observation time was a minimum of 10 min, and when the experimental bee entered the odor source bottle or stayed continuously in the 1/3 area of the bottle for 4 min, it was considered to have selected that substance. Samples and liquid paraffin in the odor source bottle were replaced after testing every five bees, maintaining the humidity. The experiment was repeated for five groups, with 10 bees featuring in each group. The olfactometer was reversed or replaced after testing two bees to eliminate interference. After each sample test, the odor source bottle and olfactometer were cleaned with a mixture of 75% alcohol and distilled water until dry, eliminating residual odors.

### 2.9. Statistical Analysis

One-way ANOVA was used to compare the gene expression of *Smo* in different tissues of honeybees and the differences in gene expression in the antennae and thorax of honeybees after drug feeding. *t*-tests were used to compare the differences in gene expression in the detection of ORs and the degree of antennal response in the EAG after drug feeding. The chi-square test was used for differences in Y-tube Olfactometer Experiments. *p* < 0.05 and *p* < 0.01 indicate statistically significant differences. The creation of statistical analysis plots was achieved using GraphPad Prism 9.5.1.

## 3. Results

### 3.1. Identification of Smo Genes

Three distinct bands, measuring 1281 bp, 1301 bp, and 1252 bp, were successfully amplified via PCR. Subsequent sequencing and assembly using DNAStar v.7.1.0 yielded the complete sequence of *Smo* (3511 bp). The online tool ORF finder predicted a coding sequence containing 2952 bp that encodes 983 amino acids. The obtained sequence was deposited in GenBank under the accession number OP616714.

### 3.2. Analysis of Differential Tissue Expression of Smo

The relative mRNA expression level of *Smo* in different tissues of *A. mellifera* was analyzed via qRT-PCR. The results showed that *Smo* was expressed in the head, thorax, abdomen, leg, wing, antennae and whole tissue, and the expression in antennae was significantly higher than that in other tissues (*p* < 0.05), followed by second higher expression in thorax and lower in head, abdomen, legs, wing and whole tissue (Figure 1).

### 3.3. Location of Smo

Immunofluorescence technology was used to explore the localization of Smo protein in the antennae, and a clear green fluorescence signal could be seen at the edge of the antennae in the antennal section of *A. mellifera*, indicating that the Smo protein was expressed in large quantities in these regions. We observed a ring of hairs at the edge of the antennae, and we speculated that the Smo might be expressed in the olfactory sensor; however, under the light microscope, we were unable to distinguish which type of olfactory sensor it was. In the blank control group, we did not add a fluorescent quencher, and we could also see that there was a clear distribution of hair-shaped sensilla at the edge of the antennae (Figure 2).

### 3.4. Effects of Cyclopamine and Purmorphamine on Smo Gene Expression

#### 3.4.1. Cyclopamine

Cyclopamine, a naturally occurring steroidal alkaloid, serves as a specific small molecule inhibitor of the Hh signaling pathway. After 24 h of cyclopamine feeding, the mRNA expression of *A. mellifera Smo* was highly significantly reduced in the 200, 500, and 800 μg/mL groups compared with those in the control group (*p* < 0.01); after extending the time to 48 h, there was no significant difference in *Smo* expression between the 200 μg/mL groups compared with that in the control group, it was significantly higher in the 500 μg/mL group compared to the control group (*p* < 0.05), and the 800 μg/mL group had significantly reduced mRNA expression compared with that in the control group (*p* < 0.05). In terms of the inhibitory effect, the 200 μg/mL group exhibited the most pronounced inhibition effect at 24 h (Figure 3a), and, thus, 200 μg/mL was selected for further functional validation.

#### 3.4.2. Purmorphamine

Purmorphamine is a small molecule agonist designed for the Smo protein and is fundamentally a purine derivative [29]. Activation of the Hh signaling pathway by purmorphamine leads to both upregulation and downregulation of downstream target genes, including *Gli1* and *ptch*. Purmorphamine binds the Smo and potently activates the Hh signaling pathway. After feeding of *A. mellifera* with purmorphamine, the 500 μg/mL group showed an increase in *Smo* mRNA expression compared to that seen in the control group after 24 h, whereas the 200 μg/mL and 800 μg/mL groups demonstrated a decrease in *Smo* mRNA expression compared to that seen in the control group. When the duration was extended to 48 h, the *Smo* mRNA expression levels increased in the 200, 500, and 800 μg/mL groups compared to those in the control group. Among these, the 800 μg/mL group exhibited the most pronounced stimulating effect at both 24 and 48 h (Figure 3b). This group was selected for further experiments.

#### 3.4.3. Regulation of OR by Smo

Based on the results of 3.2, *Smo* had the highest expression in the antennae of the *A. mellifera*, which are the olfactory organ of the honeybee and play an important role in the bee’s activities, such as foraging, so we wanted to explore whether purmorphamine and cyclopamine affects the expression of ORs. Referring to Claudianos et al.’s [28] study, we chose *OR151*, *OR152*, and *OR2* to verify our conjecture. *OR2* is an odor co-receptor. After inhibiting/activating *Smo*, the expression levels of ORs were observed. As depicted in Figure 3c, in the cyclopamine group, the expression levels of *OR152* and *OR2* significantly decreased (*p* < 0.05). In the purmorphamine group, the expression level of *OR152* significantly increased (*p* < 0.05) (Figure 3d). Thus, we speculate that changes in OR expression may be related to changes in Smo.

### 3.5. EAG Recording

The 200 μg/mL cyclopamine group and 800 μg/mL purmorphamine group were selected for the EAG test. From the results of the two groups, the antennae of *A. mellifera* responded strongly to methylheptenone and had the strongest sensitivity to this compound. In the cyclopamine group, neral had the most significant inhibitory effect (Figure 4a); in the purmorphamine group, all five standard compounds had different degrees of agonistic effect, but there was no significant difference (Figure 4b).

### 3.6. Y-Tube Olfactometer Experiments

Using the same treatment groups selected for the EAG test, five standardized compounds were tested at a concentration of 500 μg/μL, with liquid paraffin at the other end of the Y-shaped olfactometer. The reaction results are shown in Figure 5. Compared to the control group, the cyclopamine group exhibited varying degrees of inhibitory effects on the five odorants. Linalool (13%), VUAA1 (28%), methyl heptenone (58%), and neral (47%) all had selection rates significantly lower than those in the control group (*p* < 0.05) (Appendix A). Notably, VUAA1, without inhibition treatment, showed bees have the same selectivity for odorant substances as liquid paraffin; however, after cyclopamine treatment, the selection rate of liquid paraffin is higher than that of VUAA1. In the purmorphamine group, the selection rates for linalool (61%) and methyl heptenone (46%) were significantly higher than those in the control group (Appendix A).

## 4. Discussion

In the Hh pathway, Smo serves as a mediator for transducing the Hh signal across the cellular membrane [30]. Abnormal activation of Smo usually leads to basal cell carcinoma and medulloblastoma, rendering Smo a prominent therapeutic target [31]. Presently, pharmaceutical interventions, including vismodegib, sonidegib, and glasdegib, have been developed to address cancers stemming from Smo activation [32,33]. No relevant studies of Smo on *A. mellifera* have been conducted yet. In *Drosophila*, Smo protein aggregation on the cell membrane is observed upon Hh activation. The protein undergoes phosphorylation during Hh activation, facilitated by casein kinase 1 and G protein-coupled receptor kinase 2 [34]. This phosphorylation process activates the Smo protein, initiating the Hh pathway.

*Smo* exhibited the highest expression in the antennae of *A. mellifera.* Antennae are important sensory organs in insects and are capable of sensing different odors and pheromones. In insects, pheromones (and other odorants) are received via olfactory sensory neurons (OSNs) residing in cuticular structures on the antennae named sensilla. Bees have six types of olfactory sensors in their antennae: trichoid, trichoid-grooved, placoid, basiconic, coeloconic, and ampullaceum [35]. Sensilla trichodea is the most widely distributed sensor in insect antennae [36]. In honeybees, the poreplate sensilla represent the most abundant olfactory sensillum type [22]. Sensilla type, abundance and distribution in antennae of insects depend on the chemosensory need for its behaviour [37]. Immunofluorescence analysis revealed that Smo is widely distributed in the antennae of honeybees and is localized in olfactory sensors, emphasizing the close relationship between Smo and olfaction in honeybees.

Liu et al. [38] showed that the expression of *Gli* 1 mRNA, *Smo* mRNA, and Smo protein decreased after treatment with cyclopamine and that the expression of *Gli* 1 mRNA, Smo, and Smo proteins was upregulated after treatment with purmorphamine. The results of the qRT-PCR experiments of 3.2 showed that *Smo* was expressed in all tissues of *A. mellifera*, suggesting that Smo plays an important role in the life activities of *A. mellifera.* In the present study, cyclopamine produced a significant inhibitory effect on Smo, which is consistent with the results of Zhu et al. [13], who showed that cyclopamine had different inhibitory effects on Smo, PTC, and Ci in the Hh pathway; purmorphamine significantly elevated Smo expression. Purmorphamine acts directly on Smo to regulate their activities [39].

ORs play a crucial role in olfactory signal transduction and are regulated by various transcription factors, including those involved in the Hh signaling pathway [27]. We verified whether cyclopamine and purmorphamine affected OR expression. After determining the optimal inhibitory and agonistic concentrations, we measured the OR mRNA of the ORs. We found that, after cyclopamine treatment, *OR151*, *OR152*, and *OR2* showed different degrees of inhibition. After purmorphamine treatment, *OR151*, *OR152*, and *OR2* also showed agonistic effects. This was positively correlated with the changes in *Smo* expression. *Smo* knockout results in a reduced behavioral response to odors in *Drosophila*, and *Smo* indirectly regulates the expression of *Drosophila* ORs, thereby affecting their odor sensitivity [36]. To the best of our knowledge, there have been no studies on the mechanisms by which *Smo* affects *OR* expression. Therefore, we hypothesized that OR expression may be linked to changes in Smo in *A. mellifera*; however, the specific mechanism of action involved remains unclear.

In the EAG test, *A. mellifera* exhibited the strongest response to methylheptenone. In the cyclopamine group, neral showed the most significant inhibitory effect. Previous studies indicated that the optimal ligand for OR151 is linalool; OR152 showed no significant response to linalool and nerolidol but exhibited a notable affinity for neral, myrcene, and methylheptenone [28]. This influenced our selection of these odorants, with VUAA1 being the binding ligand for OR2. The amplitude of EAG responses is influenced by various factors, such as receptor quantity, the receptor response of individual OR neurons, the density of responsive neurons, and the discharge rate of neurons involved in olfaction [40]. An increase in EAG response indicates enhanced sensitivity or selective changes in the ORs. After the use of inhibitors/exciters, *A. mellifera* antennae exhibited varying degrees of inhibition/excitation in odor preferences, suggesting that cyclopamine/purmorphamine-mediated regulation of Smo may activate the Smo-Hh pathway, thereby influencing the sensitivity of ORs.

Bees sustain themselves by collecting pollen and nectar from flowers. Factors such as flower color, shape, and structure affect bee collection behavior [41], and flower volatiles are one of the main factors affecting bee collection [42]. We further validated the results using a Y-shaped olfactometer. We set a uniform concentration of 500 μg/μL in the standard compound. Following the use of cyclopamine, the most notably change in selection rate, with a 46% difference before and after inhibition, was observed for linalool compared to that seen for the control. In the purmorphamine group, the most pronounced activation effect was observed for linalool, with a difference of 28%. Behavioral experiment results were inconsistent with the findings of electrophysiological performance, a phenomenon commonly observed in insect olfactory behavior studies [43]. These variations could be attributed to changes in receptor sensitivity, receptor expression, or alterations in odor-binding proteins.

In summary, in different tissues of honey, the highest mRNA expression of *Smo* in honeybee antennae, as well as immunofluorescent Smo localized expression in the antennae and olfactory sensors, proved that Smo is closely related to honeybee olfaction. When we fed the inhibitors and agonists, the OR and Smo changes were consistent, and the EAG and behavioral tests verified the idea that Smo does affect honeybee olfaction; however, the mechanism by which Smo regulates the OR changes that affect honeybee olfaction remains unknown. The present results provide a theoretical basis for us to investigate the mechanism of the Hh signaling pathway in regulating the olfaction of honeybees.

## 5. Conclusions

*A. mellifera Smo* is most highly expressed in the antennae, and feeding affects the expression of Smo and OR, with consequent changes occurring in bee behavior. These results suggest that changes in Smo may affect the olfactory senses of honeybees. These findings provide a basis for further studies on the link between the Hh signaling pathway and olfaction in *A. mellifera*.

## Figures and Tables

**Figure 1 insects-15-00555-f001:**
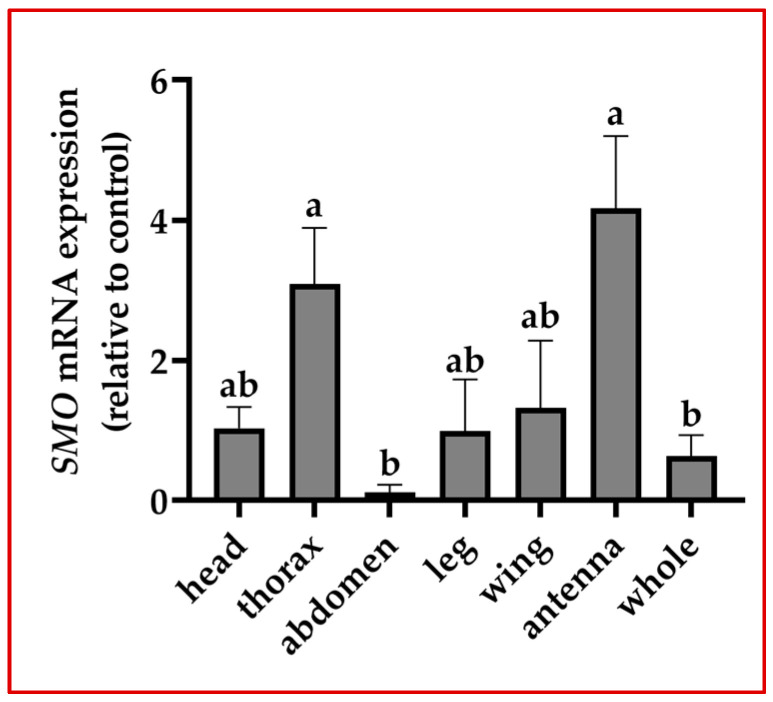
Expression of *Smo* in different tissues of *Apis mellifera*. Note: according to one-way ANOVA, the data are shown as the Mean ± SD, and the different letters represent the level of significance (*p* < 0.05).

**Figure 2 insects-15-00555-f002:**
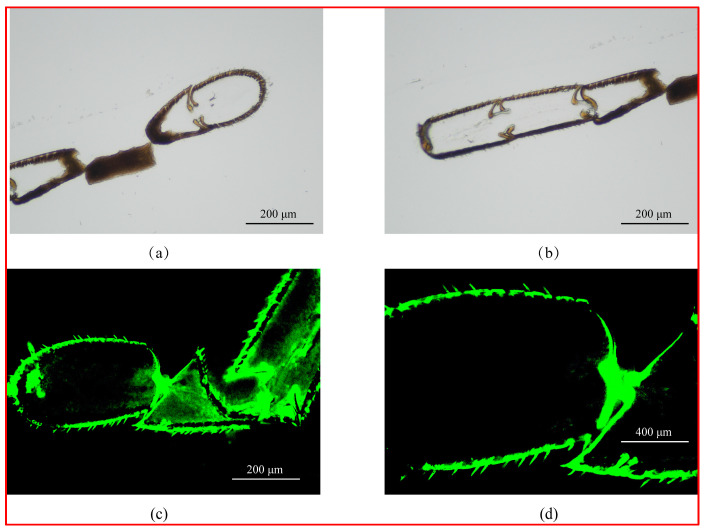
Localization of Smo in antennal tissues of *Apis mellifera*. Note: (**a**,**b**) were blank control images; bar = 200 μm. (**c**,**d**) were fluorescent photographs taken under blue excitation light. (**c**): bar = 200 µm. (**d**): bar = 400 µm.

**Figure 3 insects-15-00555-f003:**
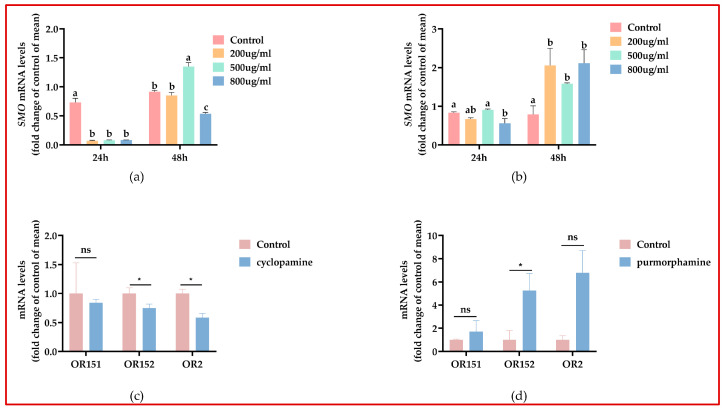
mRNA expression after treatment with cyclopamine and purmorphamine. The mRNA expression of *Smo* after treatment with different concentrations of cyclopamine (**a**) and purmorphamine (**b**); 200, 500, and 800 μg/mL were compared with the control group, respectively, and, according to one-way ANOVA, the data are shown as the Mean ± SD and the different letters represent the level of significance (*p* < 0.05). (**c**) mRNA expression of ORs after 24 h of treatment with 200 μg/mL of cyclopamine; (**d**) mRNA expression of ORs after 48 h of treatment with 800 μg/mL of purmorphamine. (**c**,**d**): Student’s *t*-tests were used to compare the differences between the drug-fed group and the control group. * *p* < 0.05. “ns” indicates no significant difference.

**Figure 4 insects-15-00555-f004:**
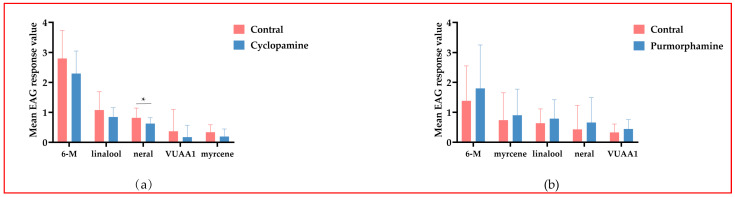
EAG responses of *Apis mellifera* to different compounds after activation/inhibition of Smo. Note: (**a**) cyclopamine; (**b**) purmorphamine; 6 m is 6-methyl-5-heptene-2-one, VUAA1 is 2-((4-Ethyl-5-(pyridin-3-yl)-4H-1,2,4-triazol-3-yl) thio)-N-(4-ethylphenyl) acetamidea. A Student’s *t*-test was used to compare the difference in EAG between the drug-fed and control groups. * *p* < 0.05.

**Figure 5 insects-15-00555-f005:**
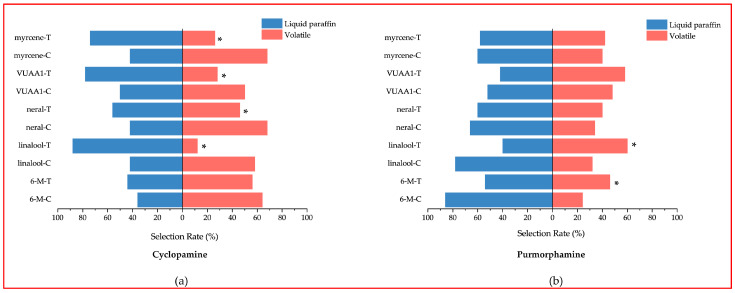
Selectivity of different odorants by *Apis mellifera* after inhibiting/activating Smo (C = 500 μg/μL). (**a**) Cyclopamine; (**b**) Purmorphamine. Each odorant is followed by a “C” for control group; “T” for treat group. Chi-square test was used to compare the difference between the drug-fed group and the liquid paraffin group, * *p* < 0.05.

**Table 1 insects-15-00555-t001:** *Smo* primer information.

Primer Name	Primer Sequence (5′-3′)	Purpose
P1	F: TCATTATTCCTAAAATTTCCCGCGT	Sequence amplification
	F: CAAATCCCTGCAGCCCAAG	
	R: TGAATGAACCGCAAATCCCTG	
P2	F: ATTTCTAGCAAATGCTTAAAGCCTT	
	R: GCACCTCTACGAGTAACTAACTTTGGTA	
P3	F: AAACATTTAATAATGCTGGTCGATT	
	F: TTAATAATGCTGGTCGATTATCTATTAGT	
	R: TTTTATATTAAAAGTTTTTCGAAAAAGAA	
*Smo*	F: TGGTGTTTGCCACTTGTCCT	qRT-PCR
	R: ACCAAGCTCTGACTGCATGA	
*Arp* *1*	F: TGCTGCACTCGTAGTTGACAATGG	
	R: ACCCTGGTGGCGTGGTCTTC	
*OR151*	F: AAACACCGGAAAGAGGTATGG	
	R: TGCATCCAGCATACGAACAG	
*OR152*	F: CGCTCTTTTGCTCTCTGGTTCG	
	R: TGATCAGCAGGCCAAGCATA	
*OR2*	F: CTCGTGGGCTCCTGTTCGCTTG	
	R: CTGTTCCTTCGGGCTGCTCTGC	

**Table 2 insects-15-00555-t002:** Standard compounds used in this study.

Compounds	CAS Number	Purity (%)	Origin
Linalool	78-70-6	>98	Macklin
Myrcene	123-35-3	>90	Macklin
6-methyl-5-heptene-2-one (6-M)	110-93-0	>98	Macklin
neral	5392-40-5	>97	Macklin
2-((4-Ethyl-5-(pyridin-3-yl)-4H-1,2,4-triazol-3-yl) thio)-N-(4-ethylphenyl) acetamidea (VUAA1)	525582-84-7	>97.5	Macklin

## Data Availability

No new data were created or analyzed in this study. Data sharing is not applicable to this article.

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
