# Peer review of "Expression and Functional Analysis of the Smo Protein in Apis mellifera"

_insects, 2024, doi:10.3390/insects15070555_

Round 1

Reviewer 1 Report (Previous Reviewer 1)

Comments and Suggestions for Authors

The authors have satisfactorily addressed my concerns and the current version of the manuscript is a significant improvement over the one previously submitted. I recommend it for publication without any change.

Comments on the Quality of English Language

 Minor

Author Response

We are so appreciated the hard work undertaken by the  reviewers on our manuscript insects-3082048. Title: Expression and functional analysis of the Smo protein in Apis mellifera. We also appreciate your previous suggestions and comments to improve our manuscripts

Reviewer 2 Report (Previous Reviewer 2)

Comments and Suggestions for Authors

              The reviewer has read the revised manuscript entitled as ‘Expression, and functional analysis of the SMO protein in Apis mellifera’ submitted by Lina Guo, Jue Wang, Diandian Yu, Yu Zhang, Huiman Zhang, Yuan Guo to INSECTS. The authors explored the expression characteristics in mainly antennal tissues of the SMO (smoothened) protein in Apis mellifera and investigated whether the inhibition or activation of SMO activity would regulate the expression of ORs using SMO antagonist and agonist in conjunction with EAG recordings and behavioral experiments. On the other hand, they deleted the description on the sequence analysis of the SMO protein in the revised manuscript and refined upon considerably the manuscript. In consequence this revised manuscript is more comprehensible than the previous manuscript. To conclude, the authors elucidated that Smo is highly expressed in antennal olfactory sensilla and hence influences on behaviors related to olfaction. However, some problems remain yet as follows.

Lines 104-106     Utilizing the GenBank reference sequence for A. mellifera Smo (XM_395373.6) and the reference gene Arp1 (GenBank accession number: NM_001185145.1), Design of primers for PCR amplification. -> (This is not a sentence. Please rewrite this part.)

Line 146              and the gas was filtered and -> (What substances did the authors use for filtering?)

Line 215              sensilla trichodea -> (How did the authors determine the sensillar type? As the sensilla are very small, it is usually difficult to distinguish sensillar types with light microscopy.)

Line 221             Figure 2 The diacritic marks (a, b, c, d) lack in Figure 2.

                            The scales in figures are too small. Please write them in large letters.

Line 253              which is an olfactory organ of the honeybee and plays an important role in the -> which are olfactory organs of the honeybee and play an important role in the

Line 255              Cyclopamine would affect -> Cyclopamine affects

Line 256              OR2 is an odor co-receptor. -> (Why did the authors select co-receptor? If the authors intend to examine the effect of the two drugs on ORx, the reviewers thinks that it is better to examine more than three ORxs. Do the authors think that only data on one ORco are less significant to elucidate its characteristics?)

Line 279              (Data deviations were shown in the first manuscript but are not shown in this revised one. Thus, the reviewer and readers cannot estimate whether or not significant differences exist between data.)

Line 281              Figure 5 -> Figure 4

Lines 297-299     Table 3 and Table 4 -> (These two tables are not referred in the text in the revised manuscript.)

Line 300             Figure5 (This figure feels wrong. The reviewer understands this experiment as follows. The authors used Y-shaped olfactometer. The odor of chemical flows from one arm and pure (control) air flows from the other arm in this olfactometer. If so, it is impossible that the attractant flows from one arm and the repellant flows from the other arm in one trail. However, Figure 5 shows it. The reviewer recommends that the result of the experiments will be presented as A / (A+B) × 100 (%).  A is the number of trails which the insect selects the arm of chemical and B is the number of trails which the insect selects the arm of pure air.)

Line 301              Selectivity of different odorants by Apis mellifera after inhibiting -> Selectivity of different odorants by Apis mellifera after inhibiting

Lines 319-320     Bees have seven types of olfactory sensors in their antennae: trichoid, trichoid-grooved, placoid, basiconic, coeloconic, campaniform and ampullaceum. -> (Campaniform sensillum is not olfactory but mechanosensory. If the description of Nation SR JL (2022) is true, the type number of olfactory sensilla on honeybee antennae is six. For reference see the following two old papers: Lacher (1964) Zeitschrift für vergleichende Physiologie 48, 587--623 and Esslen J, Kaissling K-D (1976) Zoomorphol 83:227–251)

Lines 350-351     In the cyclopamine group, neral showing the most significant inhibitory effect. -> In the cyclopamine group, neral shows the most significant inhibitory effect. (The reviewer cannot read out from Figure 5 that the neral shows the most significant inhibitory effect.)

Lines 363-375     Bees sustain themselves ……. alterations in odor-binding proteins. -> (This part of the discussion seems to be based on Figure 5. Because unfortunately the reviewer cannot understand Figure 5, the reviewer cannot understand this part of discussion.)

Line 370              neral had the highest repilling rate (60%). -> neral had the highest repelling rate (60%).

Comments on the Quality of English Language

Though there are a few careless mistakes, English is good.

Author Response

Reviewer 3 Report (Previous Reviewer 3)

Comments and Suggestions for Authors

Majors:

- authors must explain which antibodies they have used in 2.6

- Table 2 must show the abbreviations used in the text

- chapter 2.9 (Statistical Analysis) is incomplete

- line 206: expression in the thorax is not "higher" than in antennae

- Fig. 1: statistics used here must be explained in the figure legend

- Fig. 2: a, b, c and d is not shown in the figure

- numbering of the figures is wrong

- Fig. 5 (4?) does not show any statistics, although explained in the figure legend

- what does Table 3 show? Choices in percent? Then say it in the figure legend. P values shown here are not mentioned in Materials and Methods

- same is true for Table 4

- References must be written consistently.

Many typing/spelling errors and suggestions for language improvement are marked in the manuscript file.

Comments on the Quality of English Language

English needs moderate improvement.

Author Response

This manuscript is a resubmission of an earlier submission. The following is a list of the peer review reports and author responses from that submission.

Round 1

Reviewer 1 Report

Comments and Suggestions for Authors

The manuscript by Gup et al., studies the role of Smo, a component of the Hedgehog signaling pathway in the olfactory system of honeybees (Apis mellifera). The authors have well-characterized Smo in A. mellifera and also demonstrated its presence in the anetanne of honeybees. Subsequently they measured changes in the transcript level of Smo as well as two different ORs while the bees were treated with cyclopamine or purmorphamine. They also measured EAG and olfactory behavior to investigate if olfactory acuity is impacted upon by these treatments. The study is potentially an interesting one and the characterization of A.mellifera Smo is quite thorough. However, the article is very poorly written, without a well-connected coherent narrative. There are issues with several experiments and in some cases the results do not support the conclusions. 
Major Points:

1. The manuscript is very poorly written. A well-written manuscript is supposed to present a coherent narrative with interconnected sections that flow together. Instead, this one looks like discrete units, more like reports from a lab class put together. The current discussion section is mostly useless. It is mostly an amalgamation of a lot of irrelevant background information and a reiteration of the results. The authors can fuse the result and the related section from the discussion to construct a more fleshed-out result section and then totally rewrite the discussion section.

2. The reason why the particular ORs were chosen for expression analysis was not clear. Also, there is no evidence to conclude that the changes in OR transcript levels were outcomes of changes in the level of Smo transcript and not just an outcome of ingestion of the chemicals themselves. Additionally, there did not seem to be a clear pattern in which cyclopamine and purmorphamine affects Smo transcript levels, unlike what is claimed by the authors. The authors should have addressed this issue in the discussion.

3. EAG experiments were done with too few individuals. The authors need to increase the number of trials and refrain from using the result obtained from three different trials from the same antenna as three separate data points.

4. The figure legends are not sufficient and need to provide more information about the figures including but not limited to number of trials and the statistical test used.

5. The authors need to mention the posthoc test they used to make pairwise comparisons. Also, the datasets that were compared to each other need to be mentioned in the figure legend. Also 1 way ANOVA was not the correct test in several of the cases where it was used. The authors should revisit the statistics on this manuscript to fix this.

Minor Points:

1. All figure texts need to have a consistent font style and size.

2. Add suitable references for lines 62-65.

3. For section 3.1 what is the source of the template DNA for the PCR? Also, the authors need to provide more details regarding how the amplification primers were designed.

4. Expression of Smo, as shown in figure 11 is not very convincing. The authors need to provide images of the negative control so that the expression if any, can be determined. The authors should overlay these images with brightfield images and use arrows to demonstrate both the location of the cells and also, there idea of what the expression looks like. Also, the authors need to add scale bars in these images.

5. In the qPCR following the drug experiments the authors need to mention what is meant by control. The authors need to mention how. It is not clear if the authors normalized each of their data points with a control housekeeping gene. If not, the authors need to repeat these experiments, this time with control genes and should only measure relative expressions only after normalizing expression of Smo in respect to the control.

6. The authors need to mention what the error bars indicate in each of the figures. The figures should have scatter plots to show all the data points instead of just bar graphs.

7. The way the results of the T-maze assay was represented makes it very difficult to interpret. The current data may be placed into supplementary, but the authors absolutely need to present the data in form of an index so that it is more comprehensible. Also, the honey bees were observed only for 5 minutes which may be insufficient given that the bees need some time to get used to the experimental set-up before they start responding to the ambient olfactory environment. The authors need to repeat their experiments with a longer duration. In case there are precedents of previous studies using the exact same paradigm the authors may cite them to justify their experimental design.

Comments on the Quality of English Language

English in mostly fine with some minor errors which do not interfere with comprehending the manuscript. However, the way it is written is not satisfactory and the entire manuscript need to be rewritten.

Reviewer 2 Report

Comments and Suggestions for Authors

               The reviewer has read with much interest the manuscript entitled as ‘Sequence, expression, and functional analysis of the SMO protein in Apis mellifera’ submitted by Lina Guo, Jue Wang, Diandian Yu, Yu Zhang, Huiman Zhang, Yuan Guo to INSECTS. They analyzed the sequence of the Smoothened (SMO) protein in A. mellifera, explored its expression characteristics in various tissues, observed its localization expression in the antennae, and employed SMO antagonist and agonist to investigate whether the inhibition or activation of SMO activity would regulate the expression of ORs. Most parts of this research were made progress with strong supports by bioinformatics and incidentally depended on processing by software in various websites. As the reviewer cannot confirm these results, he decided to believe that the study had been adequately processed. The manuscript is fundamentally well written but some problems are present as follows.

As the names and countries of makers were not appended to some reagents and instruments used in this study, please append them.

Line 18              olfactory recognition capabilities bees remains unclear -> olfactory recognition capabilities of bees remain unclear

Lines 58-60        ORs exhibit high specificity, interacting with specific odors, and once activated, they no longer engage in broad interactions with other odors. A multifunctional receptor can bind to multiple odors, but each odor can only interact with one type of multifunctional receptor. -> Olfactory receptors are classified into two types, generalist and specialist. The generalist reacts many kinds of odors and the specialist reacts only one kind of odor such as sex pheromone receptors.

Line 113            Table 1. The three kinds of SMO primers are listed in this table. Why do three kinds of the SMO primer exist?

Line 129            Table 2. Most of the used software are listed in this table. However, some software such as DNAStar (line 221), YinOYang (line 364) and DNAMAN (Line 331) etc are not listed. How did the authors select listed software?

Line 171            and the gas was filtered -> What substances did the authors use for filtering?

Line 182            Table 3. The used compounds for odor stimulants are shown in this table. Why and how did the authors choice these compounds as adequate stimulants?

Lines 282-283    It is evident that O-glycosylation and phosphorylation at serine, threonine, and tyrosine residues, as well as SUMO modifications, may play regulatory roles in the A. mellifera SMO protein. -> The reviewer feels unusual about coinstantaneous usage of “evident” and “may” in this sentence.

Line 285            There is no explanation on the program. Please add some explanation on this script.

Line 289            http://npsa-pbil.ibcp.fr/cgi-bin/npsa_automat.pl?page=npsa_gor4.html. -> As this tool is listed in Table 2, the description is not necessary.

Line 304            in the yellow core region (Figure 5c) -> the reviewer cannot find ‘yellow core region’ in Figure 5.

Lines 310-312    Figure 5 legend. -> There are few explanations. Please add some more explanations.

Lines 345-346    Figure 9. ->It is difficult to understand this figure without explanations. Please explain the figure in more detail.

Line 350            abdomen, foot, wing, antennae and whole tissue -> abdomen, leg, wing, antennae and whole tissue

Lines 353-354    Figure 10. -> The names of some body parts are not inadequate. Chest -> thorax, belly -> abdomen, foot -> leg

Line 362            Most of these cells were in sensilla trichodea and sensilla placodea (Figure 11) -> There are several morphological types of sensilla on honeybee antennae. How did the authors discriminate these sensilla from other sensilla?

Line 365            Figure 11. -> It is difficult to discriminate each antennal part in these micrographs. Please replace these photographs with more sharp ones.

Lines 396-399    Figure 12. (a) -> Cyclopamine is effective at 24h but not at 48h. (b) Purmophamine is ineffective at 24h but effective 48h. Do these substances work in different site of SMO?  

                         (c) and (d) Cyclopamine works on OR152 and OR2 but not on OR151. Purmophamine works on OR152 but not on OR151 and OR2. What do these differences originate from?  

Line 404            A. mellifera responded -> the antennae of A. mellifera responded

Line 410            Figure 13. -> The both drugs had no effect in the EAG responses to most odors. How many did the authors record the EAG to each odor? As EAG responses are usually unstable, the reviewer thinks that it is not easy to obtain significant differences by a spatter of records.  

Lines 427-430    Figure 14. -> The results of the behavioral experiments using the Y-shaped olfactometer were shown in this figure. In the reviewer’s understanding, the test odor flowed from one arm and control air flowed from the other arm in the experiments. If so, the graphs shown in Figure 14 seem to be inappropriate.   It is better to do as follows. Initially attractive odors are selected behavioral tests and their scores (control scores) are recorded. The scores are the number of bees which run into the test arm in the fixed time. The same odors are used in the behavioral tests about agonist and antagonist to SMO and the scores are recorded (test scores). Thereafter the test scores are compared with control scores. The score differences are the larger, the regents are the more effective.

Lines 453-454    The present study identified six O-glycosylation sites, six N-glycosylation sites, 112 453phosphorylation sites, and four sumoylation sites on the SMO protein. -> The reviewer thinks that these sites were estimated from amino acid sequence but not identified.

Lines 465-473    As most of description is about other animals, it is better to shorten these.

Line 477            showed the highest expression the antennae of A. mellifera. -> showed the highest expression in the antennae of A. mellifera.

Line 482            cicadas have rich sensilla trichodea at the base -> The word ‘base’ is not good. Insect antenna consists of scape, pedicel and flagellum. Which part did the authors point between scape and pedicel?

Line 482            For example, cicadas have rich sensilla trichodea -> Why did the authors show cicadas as an example? Why did the authors not illustrate antennal sensilla of Apis as an example which were probably examined in more detail.

Lines 498-503    Activation of the Hh signaling pathway ……. effects. -> Are these discussions necessary?

Line 513            To our knowledge, to previous studies have -> To our knowledge, the previous studies have

Lines 543-549    Behavioral experiment results were …. and central brain -> If the authors think that “this process involves … and central brain”, it is better to avoid the behavioral experiments to confirm the reagent effects on cellular activity of periphery.

Line 555            SMO plays a regulatory role in modulating ORs and affecting the olfactory recognition of bees. -> Do the authors think that the later part of this sentence means that the SMO treatment in this research influences on brain function? If so, the results of behavioral experiments cannot be used for confirmation of the reagent effects on cellular activity of periphery

Reviewer 3 Report

Comments and Suggestions for Authors

Authors identified and characterized the honeybee SMO, and studied its localization and function with physiological and molecular tools. The results are important for olfactory recognition of bees, and thereby also for bee keepers. The experiments were carefully planned and carried out satisfactorily, but the manuscript needs considerable improvement.

Majors:

- Materials and methods: what is the enzyme label plate in line 94? In which L/D conditions did authors keep the bees (line 96)? Which control gene(s)  were used in 2.6? Which controls were done in the hybridization studies (2.7)? 

Results: in 3.1 the coding sequence contains 2952 bp, but in 3,2 3511. Why this discrepancy? I do not understand the legend of Fig. 1. Primers are very short and do  not have about 1000 bp! Several other figures are not adequately explained in the figure legends (Fig. 5a, Fig. 9, Fig 10, Fig 11). In 3.5, I cannot see which tissue(s) were used for the expression studies. The results described here are different to those shown in Fig. 12.

Minors:

- line 19: in bees remains

- line 37: such as bone

- line 76: localization and expression

- Fig. 3a: there is no red threshold line

- Fig, 4 and others: insert a full stop at the end of a sentence

- line 334: insert a space in the species name

- Fig. 10: do not say chest, belly and foot, but thorax, abdomen and leg (as in the text)

- Fig. 11: I miss the scale(s)

- line 372 and others: better say feeding instead of treatment

- Fig. 12:explain the significance values used in the figure legend

- line 406: insert a space before the bracket

- line 413: SMO in uppercase letters

- line 432: species name in italics

- line 477 expression in the antennae

- line 496: use a full stop instead of a comma

- line 513: delete the "to"?

Comments on the Quality of English Language

Minor language revision is necessary. I made suggestions for correction (see above).
